# Associations between Particulate Matter and Otitis Media in Children: A Meta-Analysis

**DOI:** 10.3390/ijerph17124604

**Published:** 2020-06-26

**Authors:** Sang-Youp Lee, Myoung-jin Jang, Seung Ha Oh, Jun Ho Lee, Myung-Whan Suh, Moo Kyun Park

**Affiliations:** 1Department of Otolaryngology, Wonkwang University School of Medicine, Wonkwang University Hospital, Iksan 54538, Korea; lsy738@wkuh.org; 2Medical Research Collaborating Center, Seoul National University Hospital, Seoul 03080, Korea; mjjang2014@naver.com; 3Department of Otorhinolaryngology-Head and Neck Surgery, Seoul National University College of Medicine, Seoul 03080, Korea; shaoh@snu.ac.kr (S.H.O.); junlee@snu.ac.kr (J.H.L.); drmung@naver.com (M.-W.S.); 4Sensory Organ Research Institute, Seoul National University Medical Research Center, Seoul 03080, Korea

**Keywords:** particulate matter, otitis media, child, meta-analysis

## Abstract

Particulate matter (PM), a primary component of air pollution, is a suspected risk factor for the development of otitis media (OM). However, the results of studies on the potential correlation between an increase in the concentration of PM and risk of developing OM are inconsistent. To better characterize this potential association, a meta-analysis of studies indexed in three global databases (PubMed, EMBASE, and The Cochrane Library) was conducted. These databases were systematically screened for observational studies of PM concentration and the development of OM from the time of their inception to 31 March 2020. Following these searches, 12 articles were analyzed using pooled odds ratios generated from random-effects models to test for an association between an increased concentration of PM and the risk of developing OM. The data were analyzed separately according to the size of particulate matter as PM_2.5_ and PM_10_. The pooled odds ratios for each 10 μg/m^3^ increase in PM_2.5_ and PM_10_ concentration were 1.032 (95% confidence interval (CI), 1.005–1.060) and 1.010 (95% CI, 1.008–1.012), respectively. Specifically, the pooled odds ratios were significant within the short-term studies (PM measured within 1 week of the development of OM), as 1.024 (95% CI, 1.008–1.040) for PM_2.5_ concentration and 1.010 (95% CI, 1.008–1.012) for PM_10_ concentration. They were significant for children under 2 years of age with pooled odds ratios of 1.426 (95% CI, 1.278–1.519) for an increase in the concentration of PM_2.5_. The incidence of OM was not correlated with the concentration of PM, but was correlated with an increase in the concentration of PM. In conclusion, an increase in the concentration of PM_2.5_ is more closely associated with the development of OM compared with an increase in the concentration of PM_10_; this influence is more substantial in shorter-term studies and for younger children.

## 1. Introduction

Otitis media (OM), inflammation of the middle ear, is a common disease, particularly in young children [1]. Between 1988–1994, the prevalence of OM in American children was 67–70% [2] of American children under 3 years old, 83% had at least one episode of acute OM and 46% had three or more episodes [3]. OM may:(a)Cause conductive hearing loss, otalgia, sleep disturbance, loss of appetite, and behavioral problems;(b)Delay the development of speech, language, balance, and learning abilities [4,5], thus significantly impacting the quality of life of children and their families. OM, with an estimated annual cost of 3.2 billion dollars, is one of the most costly conditions affecting children in the United States [6]. Therefore, the identification of risk factors of OM and potential ways to control for these risk factors may significantly impact global healthcare (e.g., quality of life, medical costs).

Among the many risk factors of OM, particulate matter (PM), a known risk factor for the development of asthma, bronchitis, pneumonia, and chronic obstructive pulmonary disease, has been reviewed and discussed often [7,8,9,10]. Evidence supporting an association between a change in PM exposure and the development of OM is accumulating, but inconsistent [8,11,12]. To date, no study has systematically analyzed the effect of changes in PM exposure on the development of OM in children and no consensus on this potential association has been reached. Although several review articles have analyzed the effects of air pollution as defined as total suspended particulates, NO_2_ concentration, SO_2_ concentration, and environmental tobacco smoke, the individual effects of PM were not analyzed [13,14,15,16]. Therefore, we performed a meta-analysis of studies characterizing the potential relationship between PM concentration and the incidence of OM in children.

## 2. Materials and Methods

### 2.1. Search Strategy

Searches for observational studies of OM and PM were conducted in PubMed, EMBASE, and The Cochrane Library databases from the time of their inception to 31 March 2020 (Table 1). The terms “Otitis Media” AND (“Particulate Matter” OR “Air Pollution” OR “Dust”) AND (“Prevalence” OR “Incidence” OR “Morbidity” OR “Association” OR “Risk”) were used to execute searches. Two independent screeners participated in the screening process and the specific search strategy is presented in Appendix A. The titles and abstracts of selected studies were screened, and data related to research quality, characteristics, and results were extracted.

### 2.2. Eligibility Criteria

In terms of independent variables, studies that measured PM_2.5_ or PM_10_ exposure were included. PM_2.5_ exposure was defined as the concentration of particles whose diameter was less than 2.5 μm in the air, and PM_10_ exposure was defined the concentration of particles whose diameter is less than 10 μm in the air. Exposure assessment methods were evaluated by two independent reviewers. Only qualified articles which clearly define exposure methods were included. These studies primarily acquired PM exposure from public environment centers or using a standard measuring method such as a high-volume cascade particle impactor (Harvard impactor), light-absorbing carbon tapered element oscillating microbalance (TEOM), and aerosol optical depth (AOD), etc. When it was necessary to reflect the specific time and place of each participant, regressing models using temporal variation and geographical position were also utilized.

In terms of dependent variable, studies that evaluated children with OM were included. According to the 10th revision of the International Statistical Classification of Diseases and Related Health Problems (ICD-10), H65 (nonsuppurative otitis media), H66 (suppurative and unspecified OM), H67 (OM in in diseases classified elsewhere) correspond to OM. Studies including chronically immunocompromised patients or those with an anatomical deformity or cholesteatoma were excluded.

Only studies involving human subjects and which were written in English were included. Review articles which did not treat PM were excluded.

### 2.3. Assessment of Quality and the Risk of Bias

Quality markers of the included studies, (i.e., adequacy of case selection, comparability, exposure), were assessed using the Newcastle–Ottawa Scale (NOS) (Appendix A). The NOS ratings are as follows [27]: low risk of bias (7–9); high risk of bias (4–6); and very high risk of bias (0–3) (Appendix A).

### 2.4. Data Extraction

Data were independently extracted by three authors. The following domains were checked: published year, author, study period, study region, study design, age of subjects, number of subjects, type of PM, mean PM, period of PM measurement, source of OM diagnosis, incidence of OM, measure of association, and the method used to measure PM exposure. This review was conducted in accordance with the Preferred Reporting Items for Systematic Reviews and Meta-Analyses statement [28].

### 2.5. Framework of Analysis

Data were classified into PM_2.5_ and PM_10_ groups to identify whether there were some differences between the influences of particulate matter size. For each PM group, subgroup analyses were performed according to periods of PM measurement (short-term (≤1 week) and long-term (>1 week)), study design (case-control, cohort), participant age (0–2 years and >2 years), and lag (for short-term effect studies). A funnel plot and Egger test for asymmetry were applied to assess the possibility of publication bias.

### 2.6. Statistical Analysis

The potential association between PM concentration and the incidence of OM was assessed using pooled odds ratios and their 95% confidence intervals. In a study by Yao et al. (2016) [24], risk ratios were used for daily physician visit due to OM, where the mean daily physician visit rate for OM was less than 0.01%. We included their risk ratios in a pooling analysis using an odds ratio because, when frequencies of outcomes are very low, there is a very close approximation between risk ratios and odds ratios. All included studies, except for Yao et al. (2016) [24], used odds ratios, although the study by MacIntyre et al. (2011) [19] generated an odds ratio using generalized estimating equations with a logit link function in the name of odds ratios or RR.

In a study involving multiple lag times, the largest estimate was included in the pooled analysis. In a study with multiple cohorts, cohort-specific results were included in the analysis. Risk estimates per unit increase in PM concentration were reported in all included studies, with the exception of Park et al. (2018) [11], in which risk estimates of categorical exposure compared to a reference were provided. The risk estimates per unit increase in PM were obtained directly from the authors. Odds ratios were presented for 10 μg/m^3^ increases in both PM_2.5_ and PM_10_ groups.

Pooled results are presented for random-effects models using the DerSimonian and Laird method. Statistical heterogeneity across studies was assessed by χ^2^ test and the I^2^ statistic. I^2^ values of 25%, 50%, and 75% are indicative of low, moderate, and high heterogeneity, respectively [29]. When statistical heterogeneity was low, the fixed-effect model, instead of the random effects model, could be used. However, all results were generated using random-effects models regardless of what grade statistical heterogeneity was for the target groups in the included articles.

All analyses were performed using R version 3.5.1 (The R Foundation for Statistical computing, Vienna, Austria). Two-sided *p* values <0.050 were considered indicative of statistical significance.

### 2.7. Ethical Approval

Ethics approval was not necessary, as only de-identified pooled data from individual studies were analyzed.

## 3. Results

### 3.1. Search Results

Ninety-eight references were identified following a search of the electronic literature databases. After removing 44 duplicates, 54 titles and abstracts were screened. Excluded records included seven that did not involve humans and two that were published in languages other than English. An additional sixteen records were excluded after abstract review as they were clearly not subject to analysis because they were genome-based studies or studies of allergic rhinitis. A single meeting abstract (not a full-text article) was also excluded. Among the remaining 28 articles, 10 were excluded for failing to meet the eligibility criteria, most for not using the predefined assessment method such as PM as a measure of pollution (e.g., those using total suspended particles, nitrogen dioxide concentrations, sulfur dioxide concentrations, diesel exhaust particle without using PM). An article with insufficient data was also excluded. Four review articles analyzing the relationship between air pollution and OM were also examined, however, these articles did not focus specifically on particulate matter (but rather numerous air pollutants together). An article similar to other article using the same data was excluded. In the end, 12 studies were included in this meta-analysis (Figure 1).

### 3.2. Characteristics of the Included Studies

Among the 12 studies included, five were cohort studies and seven were case–control studies. The cohort studies were from Canada, China, the Netherlands, and six other countries in Western Europe. The case–control studies were from Canada, the United States, and South Korea. Four studies included children older than five years of age. Three studies evaluated outcomes using questionnaires completed by the parents. The PM_2.5_ concentration was included in 10 studies and the PM_10_ concentration in six. The combined sample size from the 12 studies was 975,865 (Table 2) and the age range of participants was 0–18 years old.

### 3.3. Association of PM_2.5_ Exposure with the Incidence of OM

For studies including PM_2.5_ (Figure 2A), the pooled odds ratio was 1.032 (95% confidence interval (CI), 1.005–1.060). This indicates that the risk of OM incidence increased 1.032-fold for each 10 μg/m^3^ increase in PM_2.5_ concentration. 

We divided studies including PM_2.5_ into short-term (PM measured ≤1 week before the development of OM) and long-term (PM measured >1 week before the development of OM) subgroups according to the period of PM measurement (Figure 2B). In the short-term subgroup, the odds ratio was 1.024 (95% CI, 1.008–1.040). In the long-term subgroup, the odds ratio was 1.199 (95% CI, 0.849–1.693). Although the odds ratio was higher in the long-term subgroup, it was statistically significant only in the short-term subgroup.

We also divided the studies including PM_2.5_ into cohort and case–control groups depending on study design (Figure 2C). For the case–control studies, the odds ratio was 1.011 (95% CI, 0.991–1.031). For the cohort studies, the odds ratio was 1.260 (95% CI, 0.992–1.600). These results were not statistically significant for either group.

A subgroup analysis by age was also performed (Figure 2D). The odds ratios for the subgroup 0–2 years of age and the subgroup >2 years of age were 1.370 (95% CI, 1.053–1.781) and 1.018 (95% CI, 0.998-1.038), respectively, revealing statistical significance for only the younger age subgroup.

### 3.4. Association of PM_10_ Exposure with the Incidence of OM 

The pooled odds ratio of studies including PM_10_ was 1.010 (95% CI, 1.008–1.012) (Figure 3A). Therefore, the risk of OM increases 1.010-fold for each 10 μg/m^3^ increase in PM_10_ concentration. 

Next, a subgroup analysis was performed according to the period of PM measurement (Figure 3B). For the short-term (≤1 week) and long-term (>1 week) subgroups, the odds ratios were 1.010 (95% CI: 1.008–1.012) and 1.003 (95% CI, 0.927–1.085), respectively, revealing statistical significance for only the short-term subgroup.

Studies including PM_10_ exposure data were then subdivided into case-control and cohort study groups depending on the study design (Figure 3C). The case-control and cohort subgroups aligned exactly with the short-term and long-term subgroups, respectively; therefore the odds ratios were identical to those presented above for the measurement time period subgroup analysis (i.e., 1.010 (95% CI: 1.008–1.012) and 1.003 (95% CI, 0.927–1.085) for the case-control and cohort subgroups, respectively).

A subgroup analysis according to age was also performed (Figure 3D). In the subgroup 0–2 years of age, the odds ratios of random-effects models were 0.994 (95% CI, 0.921–1.073). In the subgroup >2 years of age, the odds ratios of random-effects models were 1.010 (95% CI, 1.008–1.012). The results were thus only significant in >2 years of age subgroup.

### 3.5. Trends of Odds Ratio and PM Values

In the studies including PM_2.5_ and PM_10_, no trend between odds ratio and PM values was evident. There was also no correlation between the log-transformed odds ratio and the mean PM concentration (Figure 4; *p* = 0.892 for studies including PM_2.5_ and 0.917 for those including PM_10_).

### 3.6. Publication Bias

No publication bias was identified for the included studies. Using the Egger test, the *p*-values of studies including PM_2.5_ and PM_10_ were 0.673 and 0.380, respectively. The distribution of odds ratios was not asymmetric in studies including PM_2.5_ and/or PM_10_ (Figure 5).

### 3.7. Association between Lag and the Incidence of OM

The lag between PM_2.5_ exposure and the development of OM ranged from 0–7 days (Figure 6); lag was not significantly related to the development of OM. The relationship between lag in PM_10_ exposure and OM incidence could not be studied because each study used a different lag period.

## 4. Discussion

Since 2000, concern over the effect of exposure to PM on health has grown. The WHO defines PM as a class I carcinogen, and a global exposure mortality model estimated that 8.9 million deaths are related to exposure to PM_2.5_ [30]. At the same time, air pollution is also considered a risk factor for the development of OM. Other pollutants (e.g., NO_2_, SO_2_) have already been proposed to be significant risk factors for OM [13,16,17,31]. Importantly, however, studies on the potential relationship between PM and the development of OM have yielded inconsistent results. 

Pre-clinical studies have confirmed that PM can induce the development of OM. First, an in vitro analysis revealed that PM influences the development of OM by promoting apoptosis, the expression of inflammatory cytokines (TNF-α and COX-2), and the expression of a mucin gene (Muc5AC) [12]. Furthermore, an in vivo analysis indicated that injection of PM into the middle ear of animals increases the thickness of the middle-ear mucosa and infiltration of inflammatory cells [32]. Additionally, the expression of epithelial sodium channels, which are essential for maintaining a fluid-free airway lumen, is decreased after PM exposure [32]. A transcriptomic analysis of mice exposed to diesel exhaust particles highlighted that genes related to IL-2 expression and T-cell maturation were upregulated, while CD4, IFNA1, and ESR1 were downregulated [33].

The odds ratios for the development of OM based on exposure to PM_2.5_ and PM_10_ are statistically significant (1.032 and 1.010, respectively), meaning that increased exposure to PM_2.5_ or PM_10_ can promote the development of OM. Although the odds ratio is only slightly over 1.0, the influence on society is great because:(a)the prevalence of OM is high(b)everyone is exposed to PM. It is also of note that the odds ratio is larger for increased exposure to PM_2.5_ compared with increased exposure to PM_10_; this observation is consistent with the widely accepted notion that PM_2.5_ is more toxic than PM_10_, likely because PM_2.5_ can be more easily transported to the lungs and circulatory system, and interact with mucosa and immune cells in the middle ear.

On the other hand, the influence of an “increase” in the concentration of PM and the actual concentration requires distinction. While it seems highly plausible that there would be a positive correlation between PM concentration and OM incidence, the results of this study reveal no statistically significant correlation between PM concentration and odds ratio for the development of OM (Figure 4). Although it is clear that the incidence of OM is impacted by increases in the concentration of PM, there is no direct evidence that its incidence is influenced by the actual concentration of PM. However, more studies are required to better understand the associations between concentrations of PM and OM and the potential regional effects, particularly in regions with very high concentrations of PM (e.g., China, South Korea), since higher odds ratios in these areas have recently been reported.

Studies included in this meta-analysis were divided into “short-term” and “long-term” subgroups according to the period of PM concentration measurement. When a study analyzed the concentration of PM for less than one week before the occurrence of OM, it was classified as a “short-term” study. This type of study might assume that OM was primarily influenced by recent changes in the concentration of PM. By contrast, a “long-term” study is one that included assessments of PM concentration for more than one week. However, it is important to note that the measurement period for all long-term studies included here was ≥60 days; there were no studies whose measurement period was between 1 week and 2 months. Long-term studies might assume that OM occurred as a result of cumulative PM exposure. Therefore, it was very difficult to measure lag period accurately for long-term studies, and the averaging period was used as the best alternative estimate of lag period. In this analysis, the incidence of OM was affected by short-term increases in PM_2.5_ or PM_10_ concentration, a result supported by the observation that most pediatric cases of OM are acute.

Young children are highly susceptible to the deleterious effects of environmental pollution and it is widely known that OM is very common in very young children. Importantly, however, the definition of ‘very young’ varies according to the researchers. We considered 2 or 3 as the age criteria simultaneously, and chose 2 because the study results were more clearly differentiated around this age. Interestingly, in this study, an increase in exposure to PM_10_ led to an increased incidence of OM in children >2 years of age, while an increase in exposure to PM_2.5_ led to an increased incidence of OM in children 0–2 years of age. Importantly, the PM_10_ results were largely influenced by the study by Park et al. (2018) [11], which only included children >5 years of age—data that are not consistent with other studies included here. Because this may be a major source of bias, it is not possible to affirm this finding; follow-up studies are highly recommended.

The potential influence of lag (i.e., time between exposure to PM and onset of OM) was also analyzed. In the group of studies including PM_2.5_, lags of 0–7 days did not significantly influence the incidence of OM (Figure 6). In this subgroup, however, there were some differences in results between studies. For instance, while Zemek et al. (2010) [18] reported the lowest odds ratio and Kousha et al. (2016) [21] reported the highest odds ratio, both included the same lag time (i.e., 3 days after exposure to PM_2.5_). Zemek et al. (2010) [18] also assessed 0–4-day lags after PM_10_ exposure but did not note any significant differences. Although Xiao et al. (2016) [23] and Park et al. (2018) [11] reported significantly positive risk ratios, their results could not be analyzed together because of different lag times. In the group of studies including PM_10_, an analysis of each lag was impossible because the lags used in each study were different. 

In this study, two pairs of studies analyzed overlapping populations. The first pair included studies by Strickland et al. (2016) [22] and Xiao et al. (2016) [23]. Each had different PM measuring methods (MAIAC aerosol optical depth and two-stage spatiotemporal model vs. CMAQ model simulations and ground-based measurements) and periods (1~2 days vs. 3 days). The second pair included studies by Girguis et al. (2017) [26] and Girguis et al. (2018) [8]. Each of these studies also used different methods to measure PM (three-stage statistical model vs. model using satellite) and period (lifetime vs. 0~7 days). Although they had overlapping participants, each study had unique measuring methods and periods. Therefore, we included these studies and regarded them as separate studies.

This study has several limitations. First, the methods used to estimate the concentration of PM could be problematic; some studies used monitoring stations while others used their own modeling method using external data (e.g., traffic information, weather records, distance from a subjects’ residence to the road). Second, the results of questionnaires may be influenced by recall bias. In this study, three studies which used questionnaires report higher odds ratios compared with other studies. The influence of recall bias in these studies could not be estimated and excluded. Finally, there may have also been selection bias in the included studies (e.g., challenges visiting the emergency room for low-income families or those in rural areas).

## 5. Conclusions

Increases in the concentration of PM are weakly correlated with the development of OM in children. An increase in the concentration of PM_2.5_ was better correlated with the development of OM compared with an increase in the concentration of PM_10_. Furthermore, the development of OM was correlated with only a short-term increase in PM concentration. Children 0–2 years of age were more vulnerable to PM exposure compared with those older than 2 years of age. No correlation between lag in exposure to PM and development of OM was identified.

## Figures and Tables

**Figure 1 ijerph-17-04604-f001:**
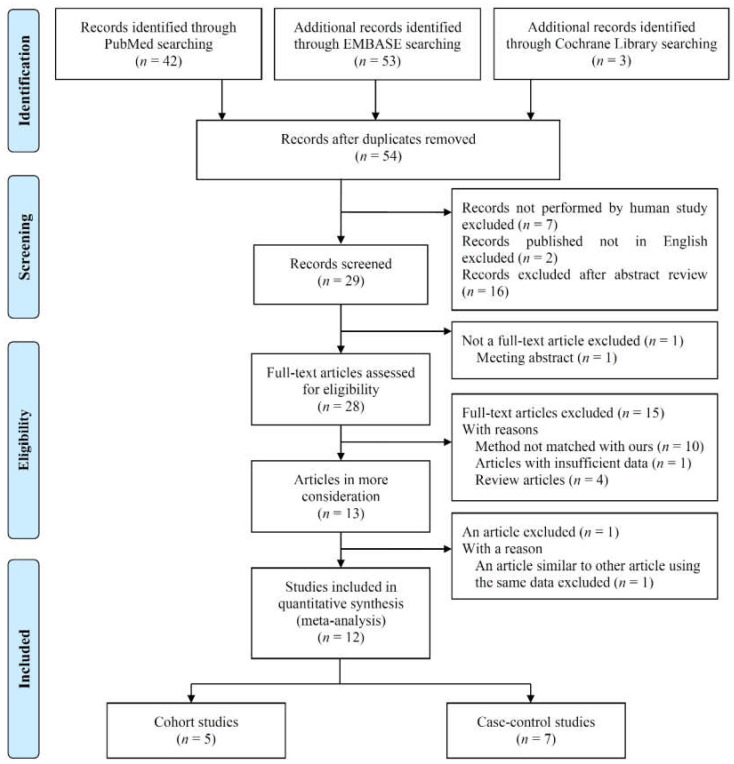
PRISMA 2009 flow diagram of the study selection process for this meta-analysis.

**Figure 2 ijerph-17-04604-f002:**
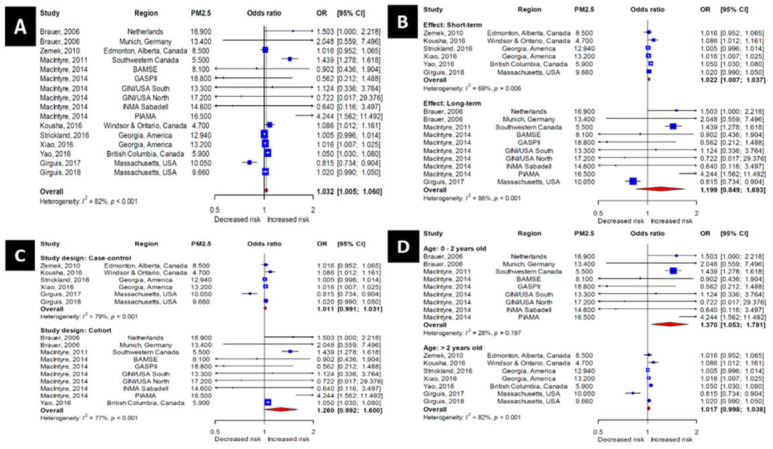
Forest plots for the PM_2.5_ group. (**A**)—All subjects. (**B**)—Short- and long-term effect models. (**C**)—Case–control and cohort studies. (**D**)—Younger and older children.

**Figure 3 ijerph-17-04604-f003:**
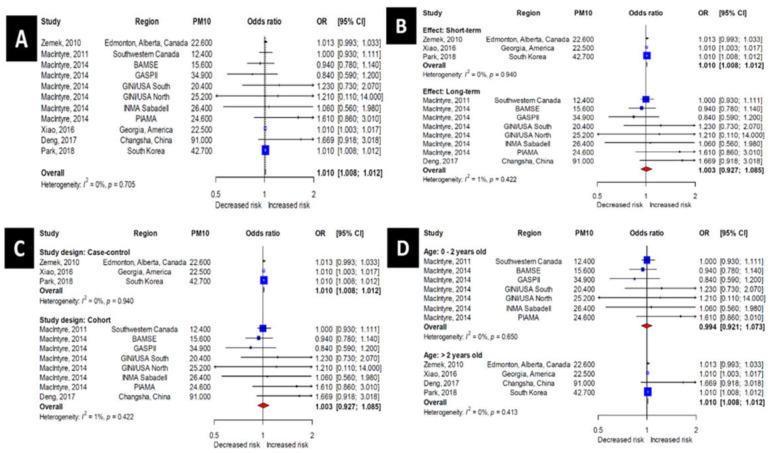
Forest plots for the PM_10_ group. (**A**)—All subjects. (**B**)—Short- and long-term effect models. (**C**)—Case–control and cohort studies. (**D**)—Younger and older children.

**Figure 4 ijerph-17-04604-f004:**
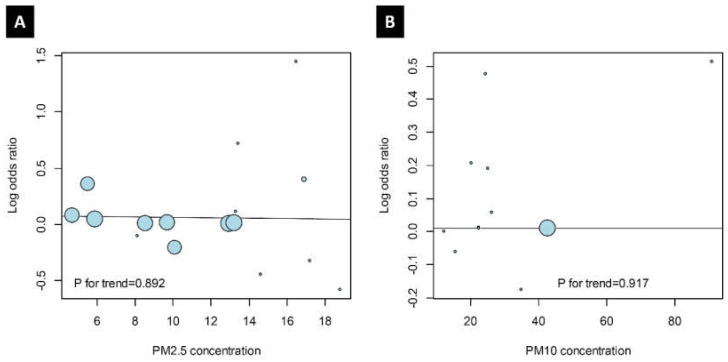
Trends between odds ratios and representative PM values in the (**A**) PM_2.5_ and (**B**) PM_10_ groups. Each circle is a study included in this article, and its area means the number of its subjects.

**Figure 5 ijerph-17-04604-f005:**
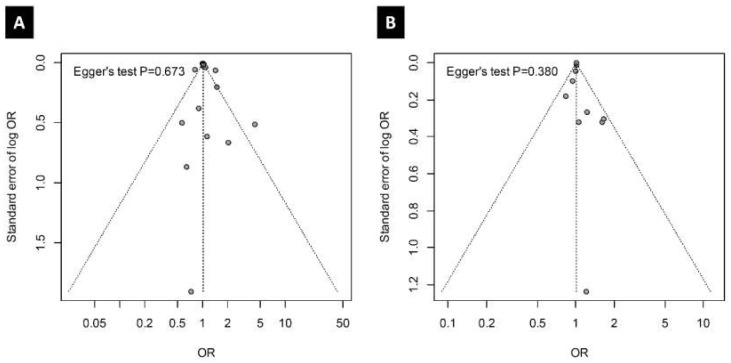
Funnel plot of publication bias in the (**A**)—PM_2.5_ and (**B**)—PM_10_ groups.

**Figure 6 ijerph-17-04604-f006:**
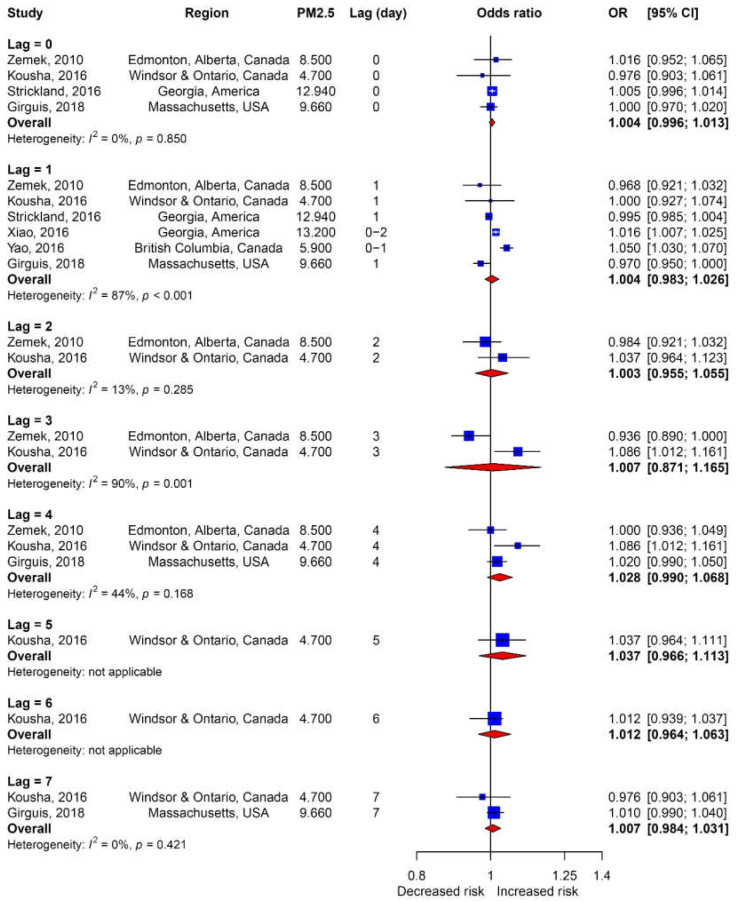
Association of lag with the incidence of OM.

**Table 1 ijerph-17-04604-t001:** Characteristics of included studies.

Number	Published Year	Author	Study Period	Study Region	Study Design	OM Diagnosis Source	Age of Subjects (yr)	Number of Subjects	Type of PM	Mean PM	Period of PM Measurement (d)	Classification of PM Measurement Period *	Measure of Association	Per Increase	Study Quality
1	2006	Brauer [17]	2002–2003	Netherlands	Cohort study	Parent report	0–1	2984	PM_2.5_	16.9	365, 730	Long–term	Odds ratio	3	7
2	2010	Zemek [18]	1992–2002	Edmonton, Alberta, Canada	Case-control study	Medical record	1–3	14,527	PM_10_	22.6	0–4	Short-term	Odds ratio	15	7
3	2011	MacIntyre [19]	1999–2000	Southwestern Canada	Cohort study	Medical record	0–2	44,917	PM_10_	12.4	60	Long-term	Odds ratio	2.8	8
4	2014	MacIntyre [20]	2008–2011	Six Countries in Western Europe	Cohort study	Parent report	0	8772	PM_2.5_	8.1–18.8	365	Long-term	Odds ratio	5	8
5	2016	Kousha [21]	2004–2010	Windsor & Ontario, Canada	Case-control study	Medical record	0–3	4815	PM_2.5_	4.7	0–7	Short-term	Odds ratio	8.2	8
6	2016	Strickland [22]	2002–2010	Georgia, America	Case-control study	Medical record	0–18	237,833	PM_2.5_	12.94	1–2	Short-term	Odds ratio	10	7
7	2016	Xiao [23]	2002–2008	Georgia, America	Case–control study	Medical record	0–18	422,268	PM_10_	22.5	3	Short-term	Odds ratio	11.5	7
8	2016	Yao [24]	2003–2010	British Columbia, Canada	Cohort study	Medical record	0–10	175	PM_2.5_	5.9	1	Short-term	Risk ratio	10	8
9	2017	Deng [25]	2011–2012	Changsha, China	Cohort study	Parent report	0	1617	PM_10_	106	90–1095	Long-term	Odds ratio	15	7
10	2017	Girguis [26]	2001–2006	Massachusetts, USA	Case-control study	Medical record	0–3	40,042	PM_2.5_	10.1	0–1095	Long-term	Odds ratio	2	8
11	2018	Girguis [8]	2001–2008	Massachusetts, USA	Case-control study	Medical record	0–3	37,040	PM_2.5_	9.56–9.76	0–7	Short-term	Odds ratio	10	8
12	2018	Park [11]	2011–2012	South Korea	Case-control study	Medical record	0–14	160,875	PM_10_	42.7	7	Short-term	Odds ratio	30	8

Note: ^*^—Short-term: Particulate matter (PM) concentration measured ≤1 week from occurrence of otitis media (OM); long-term: PM concentrations measured ≥1 week from occurrence of OM.

**Table 2 ijerph-17-04604-t002:** Comparative sample size of each subgroup.

Study	Group	Subgroup	
Overall 975,865	PM_2.5_ 813,181	Short-term 716,708	Long-term 96,473
		Case-control 756,575	Cohort 56,606
		<3 years old 56,431	≥3 years old 756,750
	PM_10_ 653,942	Short-term 597,670	Long-term 56,272
		Case-control 597,670	Cohort 56,272
		<3 years old 54,655	≥3 years old 599,287

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
