# Peer review of "Associations between Particulate Matter and Otitis Media in Children: A Meta-Analysis"

_ijerph, 2020, doi:10.3390/ijerph17124604_

Round 1

Reviewer 1 Report

This manuscript is interesting, well structured, robust data supports the conclusions. The authors provided an evidence of influence of air pollution on the otitis media in children in world literature cross section. The information is valuable, and I guess it will find a wide range of readers. A high portion of delicious paper and statistical work has been done.

However, that information is already known. The pollution of the air influences the occurrence of otitis media in small children. This topic was studied and published more than 30 years ago. Despite this I have no hesitation to recommend this article in this version.  The only thing I had the problem to clearly understand was the division of particles to two subgroups, this should be mentioned even in abstract and more clearly elucidated.

Author Response

Response to Reviewer 1 Comments

This manuscript is interesting, well structured, robust data supports the conclusions. The authors provided an evidence of influence of air pollution on the otitis media in children in world literature cross section. The information is valuable, and I guess it will find a wide range of readers. A high portion of delicious paper and statistical work has been done.

However, that information is already known. The pollution of the air influences the occurrence of otitis media in small children. This topic was studied and published more than 30 years ago. Despite this I have no hesitation to recommend this article in this version.

Response: Thank you for your kind comments. Although the overall influence of air pollution to otitis media is already known, the influence of particulate matter has not been clearly known. Considering the high interest of particulate matter, we think it is very important to investigate how otitis media would be influenced by particulate matter quantitatively.

The only thing I had the problem to clearly understand was the division of particles to two subgroups, this should be mentioned even in abstract and more clearly elucidated.

Response: The division of particles into PM10 and PM2.5 is common to studies of particulate matter. We commented about that in abstract in line 24-25 and methods in line 107-108.

Reviewer 2 Report

I thank the editors for the opportunity to review this article titled Associations between Particulate Matter and Otitis Media in Children: A Meta-Analysis. In this article, the authors report a challenging and well-designed systematic review and meta-analysis into the impact of different sizes of particulate matter in the air (pollution, dust) on middle-ear inflammation in infants.

The article introduces the economic and health impacts of otitis media and the possible association with particulate matter. The authors fully describe the systematic review methodology, inclusion and exclusion criteria and assessment of bias for contributing articles.

Below are some comments that the authors may wish to integrate into future revisions of their work:

My only major concerns are around the justification of the analysis choices made. The article introduces OM as having the greatest impact in children under 3, yet the framework of analysis (lines 104-109) section divides the data into children aged 0 to 2, and > 2 years. So by that assumption, a child aged 2 and a half would fall into the older group? Please could the threshold be clarified if this is not the case, and justified? Further is the definition of “acute” widely accepted to be ≤ 1 week? Is there a justification for this?

A second question I have is around the sample sizes of the studies involved. The overall sample size of 975,865 is very large – it would be useful for the reader to have some breakdown of how this sample size is distributed across – specifically, comparative subgroup sizes in the subgroup analyses. Please could this be added into the text and/or figures (even just added to Appendix Table B2)?

Further comments in the format of line-by-line instruction:

Figure 2 is very difficult to read and interpret at the size and reproduction quality given. Please either make the subfigures larger or improve the reproduction quality. Figure 6 is of much more appropriate size and clarity.

There are a few typos and/or English language errors in the article e.g. line 84 “independent variable” should be “independent variables”.

Author Response

Response to Reviewer 2 Comments

I thank the editors for the opportunity to review this article titled Associations between Particulate Matter and Otitis Media in Children: A Meta-Analysis. In this article, the authors report a challenging and well-designed systematic review and meta-analysis into the impact of different sizes of particulate matter in the air (pollution, dust) on middle-ear inflammation in infants.

The article introduces the economic and health impacts of otitis media and the possible association with particulate matter. The authors fully describe the systematic review methodology, inclusion and exclusion criteria and assessment of bias for contributing articles.

Below are some comments that the authors may wish to integrate into future revisions of their work:

My only major concerns are around the justification of the analysis choices made. The article introduces OM as having the greatest impact in children under 3, yet the framework of analysis (lines 104-109) section divides the data into children aged 0 to 2, and > 2 years. So by that assumption, a child aged 2 and a half would fall into the older group? Please could the threshold be clarified if this is not the case, and justified? Further is the definition of “acute” widely accepted to be ≤ 1 week? Is there a justification for this?

Response: Thank you for your considerate pointing out. Because ‘children under 3’ means children aged 0 to 3, it is different from the criteria of this article. It seems that there are no unified criteria how to divide children age about otitis media. In pediatrics, infancy and early childhood are differentiated around age of 2 years. A child aged 2 and a half is fall into the older group. We clarified this by using consistent criteria in the line 183, 202-203, 285, and 320.

In addition, the definition of “acute” in “acute otitis media” is generally accepted to be “occurred within 3 weeks”. Contrarily, there is no unified definition of “short” in “short-term”, and we defined “short-term” as “within a week”. In our manuscript, we stated “In this analysis, the incidence of OM was affected by short-term increases in PM2.5 or PM10 concentration, a result supported by the observation that most pediatric cases of OM are acute” in line 276-278. Because we define “short-term” as “within a week”, it is true that most pediatric cases of OM are acute (occurred within 3 weeks) according to the results of our analysis.

A second question I have is around the sample sizes of the studies involved. The overall sample size of 975,865 is very large – it would be useful for the reader to have some breakdown of how this sample size is distributed across – specifically, comparative subgroup sizes in the subgroup analyses. Please could this be added into the text and/or figures (even just added to Appendix Table B2)?

Response: We added a table (Table 2) which shows how the sample size was distributed (line 166-167).

Further comments in the format of line-by-line instruction:

Figure 2 is very difficult to read and interpret at the size and reproduction quality given. Please either make the subfigures larger or improve the reproduction quality. Figure 6 is of much more appropriate size and clarity.

Response: We prepared each subfigure enough resolution (about 4,000*2,500). We will ask the editor to print these figures horizontally or to arrange each subfigure vertically one by one.

There are a few typos and/or English language errors in the article e.g. line 84 “independent variable” should be “independent variables”.

Response: We fixed that typo properly (line 76).

Reviewer 3 Report

This manuscript reports the results of a meta-analysis of studies characterizing the potential relationship between the concentration of particulate matter (PM) and the incidence of otitis media (OM) in children. The analysis addressed an important topic in public health, because OM is highly prevalent in young children. Twelve studies met the inclusion criteria, although unavoidably there were procedural differences across those studies. The overall finding was that there is a weak association between increases in PM concentration and increased OM. The manuscript provides a thorough discussion, that considers both the importance of the findings and the limitations of the approach.  The topic is clearly appropriate for this journal.

Minor comments:

line 19: These databases do not "publish" papers; "indexed in ... databases" would be more accurate.

line 151: Extra word in the sentence that starts "And An"

figure 4: What do the different symbols mean?

Author Response

Response to Reviewer 3 Comments

This manuscript reports the results of a meta-analysis of studies characterizing the potential relationship between the concentration of particulate matter (PM) and the incidence of otitis media (OM) in children. The analysis addressed an important topic in public health, because OM is highly prevalent in young children. Twelve studies met the inclusion criteria, although unavoidably there were procedural differences across those studies. The overall finding was that there is a weak association between increases in PM concentration and increased OM. The manuscript provides a thorough discussion, that considers both the importance of the findings and the limitations of the approach. The topic is clearly appropriate for this journal.

Minor comments:

line 19: These databases do not "publish" papers; "indexed in ... databases" would be more accurate.

Response: Thank you for your delicate pointing out. We fixed that expression as you pointed out in line 19.

line 151: Extra word in the sentence that starts "And An"

Response: We removed “And” in the line 154.

figure 4: What do the different symbols mean?

Response: We added the explanation what those symbols mean in the caption of figure 4.